# Supervised Learning with Tensor Networks

**E. M. Stoudenmire**
Perimeter Institute for Theoretical Physics
Waterloo, Ontario, N2L 2Y5, Canada

**David J. Schwab**
Department of Physics
Northwestern University, Evanston, IL

## Abstract

Tensor networks are approximations of high-order tensors which are efficient to work with and have been very successful for physics and mathematics applications. We demonstrate how algorithms for optimizing tensor networks can be adapted to supervised learning tasks by using matrix product states (tensor trains) to parameterize non-linear kernel learning models. For the MNIST data set we obtain less than $1\%$ test set classification error. We discuss an interpretation of the additional structure imparted by the tensor network to the learned model.

## 1 Introduction

Recently there has been growing appreciation for tensor methods in machine learning. Tensor decompositions can solve non-convex optimization problems [1, 2] and be used for other important tasks such as extracting features from input data and parameterizing neural nets [3, 4, 5]. Tensor methods have also become prominent in the field of physics, especially the use of *tensor networks* which accurately capture very high-order tensors while avoiding the the curse of dimensionality through a particular geometry of low-order contracted tensors [6]. The most successful use of tensor networks in physics has been to approximate exponentially large vectors arising in quantum mechanics [7, 8].

Another context where very large vectors arise is non-linear kernel learning, where input vectors $\mathbf{x}$ are mapped into a higher dimensional space via a feature map $\Phi(\mathbf{x})$ before being classified by a decision function

$$f(\mathbf{x}) = W \cdot \Phi(\mathbf{x}) . \qquad (1)$$

The feature vector $\Phi(\mathbf{x})$ and weight vector $W$ can be exponentially large or even infinite. One approach to deal with such large vectors is the well-known kernel trick, which only requires working with scalar products of feature vectors [9].

In what follows we propose a rather different approach. For certain learning tasks and a specific class of feature map $\Phi$, we find the optimal weight vector $W$ can be approximated as a tensor network—a contracted sequence of low-order tensors. Representing $W$ as a tensor network and optimizing it directly (without passing to the dual representation) has many interesting consequences. Training the model scales only linearly in the training set size; the evaluation cost for a test input is independent of training set size. Tensor networks are also adaptive: dimensions of tensor indices internal to the network grow and shrink during training to concentrate resources on the particular correlations within the data most useful for learning. The tensor network form of $W$ presents opportunities to extract information hidden within the trained model and to accelerate training by optimizing different internal tensors in parallel [10]. Finally, the tensor network form is an additional type of regularization beyond the choice of feature map, and could have interesting consequences for generalization.

One of the best understood types of tensor networks is the matrix product state (MPS) [11, 8], also known as the tensor train decomposition [12]. Though MPS are best at capturing one-dimensional correlations, they are powerful enough to be applied to distributions with higher-dimensional correlations as well. MPS have been very useful for studying quantum systems, and have recently

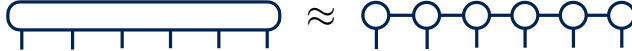

Figure 1: The matrix product state (MPS) decomposition, also known as a tensor train. (Lines represent tensor indices and connecting two lines implies summation.)

been investigated for machine learning applications such as learning features by decomposing tensor representations of data [4] and compressing the weight layers of neural networks [5].

While applications of MPS to machine learning have been a success, one aim of the present work is to have tensor networks play a more central role in developing learning models; another is to more easily incorporate powerful algorithms and tensor networks which generalize MPS developed by the physics community for studying higher dimensional and critical systems [13, 14, 15]. But in what follows, we only consider the case of MPS tensor networks as a proof of principle.

The MPS decomposition is an approximation of an order-N tensor by a contracted chain of N lower-order tensors shown in Fig. 1. (Throughout we will use tensor diagram notation: shapes represent tensors and lines emanating from them are tensor indices; connecting two lines implies contraction of a pair of indices. We emphasize that tensor diagrams are not merely schematic, but have a rigorous algorithmic interpretation. For a helpful review of this notation, see Cichocki [16].)

Representing the weights $W$ of Eq. (1) as an MPS allows one to efficiently optimize these weights and adaptively change their number by varying $W$ locally a few tensors at a time, in close analogy to the density matrix renormalization group (DMRG) algorithm used in physics [17, 8]. Similar alternating least squares methods for tensor trains have been explored more recently in applied mathematics [18].

This paper is organized as follows: first we propose our general approach and describe an algorithm for optimizing the weight vector $W$ in MPS form. Then we test our approach on the MNIST handwritten digit set and find very good performance for remarkably small MPS bond dimensions. Finally, we discuss the structure of the functions realized by our proposed models.

For researchers interested in reproducing our results, we have made our codes publicly available at: https://github.com/emstoudenmire/TNML. The codes are based on the ITensor library [19].

## 2    Encoding Input Data

Tensor networks in physics are typically used in a context where combining $N$ independent systems corresponds to taking a tensor product of a vector describing each system. With the goal of applying similar tensor networks to machine learning, we choose a feature map of the form

$$\Phi^{s_1 s_2 \cdots s_N}(\mathbf{x}) = \phi^{s_1}(x_1) \otimes \phi^{s_2}(x_2) \otimes \cdots \phi^{s_N}(x_N) . \tag{2}$$

The tensor $\Phi^{s_1 s_2 \cdots s_N}$ is the tensor product of a local feature map $\phi^{s_j}(x_j)$ applied to each input component $x_j$ of the $N$-dimensional vector $\mathbf{x}$ (where $j = 1, 2, \ldots, N$). The indices $s_j$ run from 1 to $d$, where $d$ is known as the local dimension and is a hyper-parameter defining the classification model. Though one could use a different local feature map for each input component $x_j$, we will only consider the case of homogeneous inputs with the same local map applied to each $x_j$. Thus each $x_j$ is mapped to a $d$-dimensional vector, and the full feature map $\Phi(\mathbf{x})$ can be viewed as a vector in a $d^N$-dimensional space or as an order-$N$ tensor. The tensor diagram for $\Phi(\mathbf{x})$ is shown in Fig. 2. This type of tensor is said be rank-1 since it is manifestly the product of $N$ order-1 tensors.

For a concrete example of this type of feature map, which we will use later, consider inputs which are grayscale images with $N$ pixels, where each pixel value ranges from 0.0 for white to 1.0 for black. If the grayscale value of pixel number $j$ is $x_j \in [0, 1]$, a simple choice for the local map $\phi^{s_j}(x_j)$ is

$$\phi^{s_j}(x_j) = \left[\cos\left(\frac{\pi}{2}x_j\right), \ \sin\left(\frac{\pi}{2}x_j\right)\right] \tag{3}$$

and is illustrated in Fig. 3. The full image is represented as a tensor product of these local vectors. The above feature map is somewhat ad-hoc, and is motivated by "spin" vectors encountered in quantum systems. More research is needed to understand the best choices for $\phi^s(x)$, but the most crucial property seems to be that $\vec{\phi}(x) \cdot \vec{\phi}(x')$ is a smooth and slowly varying function of $x$ and $x'$, and induces a distance metric in feature space that tends to cluster similar images together.

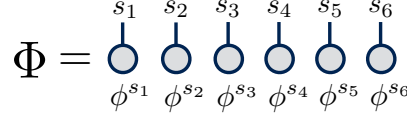

Figure 2: Input data is mapped to a normalized order N tensor with a rank-1 product structure.

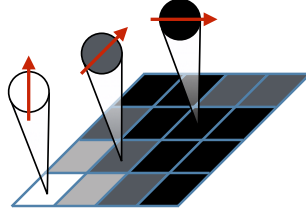

Figure 3: For the case of a grayscale image and $d = 2$, each pixel value is mapped to a normalized two-component vector. The full image is mapped to the tensor product of all the local pixel vectors as shown in Fig. 2.

The feature map Eq. (2) defines a kernel which is the product of $N$ local kernels, one for each component $x_j$ of the input data. Kernels of this type have been discussed previously in Vapnik [20, p. 193] and have been argued by Waegeman et al. [21] to be useful for data where no relationship is assumed between different components of the input vector prior to learning.

## 3   Classification Model

In what follows we are interested in classifying data with pre-assigned hidden labels, for which we choose a "one-versus-all" strategy, which we take to mean optimizing a set of functions indexed by a label $\ell$

$$f^\ell(\mathbf{x}) = W^\ell \cdot \Phi(\mathbf{x}) \tag{4}$$

and classifying an input $\mathbf{x}$ by choosing the label $\ell$ for which $|f^\ell(\mathbf{x})|$ is largest.

Since we apply the same feature map $\Phi$ to all input data, the only quantity that depends on the label $\ell$ is the weight vector $W^\ell$. Though one can view $W^\ell$ as a collection of vectors labeled by $\ell$, we will prefer to view $W^\ell$ as an order $N + 1$ tensor where $\ell$ is a tensor index and $f^\ell(\mathbf{x})$ is a function mapping inputs to the space of labels. The tensor diagram for evaluating $f^\ell(\mathbf{x})$ for a particular input is depicted in Fig. 4.

Because the weight tensor $W^\ell_{s_1 s_2 \cdots s_N}$ has $N_L \cdot d^N$ components, where $N_L$ is the number of labels, we need a way to regularize and optimize this tensor efficiently. The strategy we will use is to represent $W^\ell$ as a *tensor network*, namely as an MPS which have the key advantage that methods for manipulating and optimizing them are well understood and highly efficient. An MPS decomposition of the weight tensor $W^\ell$ has the form

$$W^\ell_{s_1 s_2 \cdots s_N} = \sum_{\{\alpha\}} A^{\alpha_1}_{s_1} A^{\alpha_1 \alpha_2}_{s_2} \cdots A^{\ell; \alpha_j \alpha_{j+1}}_{s_j} \cdots A^{\alpha_{N-1}}_{s_N} \tag{5}$$

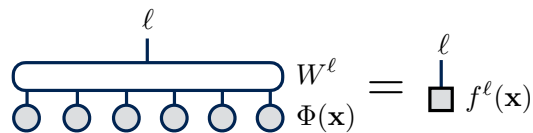

Figure 4: The overlap of the weight tensor $W^\ell$ with a specific input vector $\Phi(\mathbf{x})$ defines the decision function $f^\ell(\mathbf{x})$. The label $\ell$ for which $f^\ell(\mathbf{x})$ has maximum magnitude is the predicted label for $\mathbf{x}$.

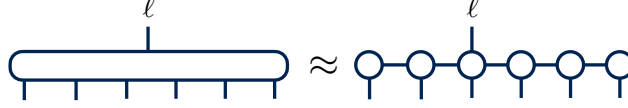

Figure 5: Approximation of the weight tensor $W^\ell$ by a matrix product state. The label index $\ell$ is placed arbitrarily on one of the $N$ tensors but can be moved to other locations.

and is illustrated in Fig. 5. Each $A$ tensor has $d\,m^2$ elements which are the latent variables parameterizing the approximation of $W$; the $A$ tensors are in general not unique and can be constrained to bestow nice properties on the MPS, like making the $A$ tensors partial isometries.

The dimensions of each internal index $\alpha_j$ of an MPS are known as the *bond dimensions* and are the (hyper) parameters controlling complexity of the MPS approximation. For sufficiently large bond dimensions an MPS can represent any tensor [22]. The name matrix product state refers to the fact that any specific component of the full tensor $W^\ell_{s_1 s_2 \cdots s_N}$ can be recovered efficiently by summing over the $\{\alpha_j\}$ indices from left to right via a sequence of matrix products (the term "state" refers to the original use of MPS to describe quantum states of matter).

In the above decomposition Eq. (5), the label index $\ell$ was arbitrarily placed on the tensor at some position $j$, but this index can be moved to any other tensor of the MPS without changing the overall $W^\ell$ tensor it represents. To do so, one contracts the tensor at position $j$ with one of its neighbors, then decomposes this larger tensor using a singular value decomposition such that $\ell$ now belongs to the neighboring tensor—see Fig. 7(a).

## 4 "Sweeping" Optimization Algorithm

Inspired by the very successful DMRG algorithm developed for physics applications [17, 8], here we propose a similar algorithm which "sweeps" back and forth along an MPS, iteratively minimizing the cost function defining the classification task.

To describe the algorithm in concrete terms, we wish to optimize the quadratic cost $C = \frac{1}{2} \sum_{n=1}^{N_T} \sum_\ell (f^\ell(\mathbf{x}_n) - y_n^\ell)^2$ where $n$ runs over the $N_T$ training inputs and $y_n^\ell$ is the vector of desired outputs for input $n$. If the correct label of $\mathbf{x}_n$ is $L_n$, then $y_n^{L_n} = 1$ and $y_n^\ell = 0$ for all other labels $\ell$ (i.e. a one-hot encoding).

Our strategy for minimizing this cost function will be to vary only two neighboring MPS tensors at a time within the approximation Eq. (5). We could conceivably just vary one at a time, but varying two tensors makes it simple to adaptively change the MPS bond dimension.

Say we want to improve the tensors at sites $j$ and $j+1$. Assume we have moved the label index $\ell$ to the MPS tensor at site $j$. First we combine the MPS tensors $A^\ell_{s_j}$ and $A_{s_{j+1}}$ into a single "bond tensor" $B^{\alpha_{j-1} \ell \alpha_{j+1}}_{s_j s_{j+1}}$ by contracting over the index $\alpha_j$ as shown in Fig. 6(a).

Next we compute the derivative of the cost function $C$ with respect to the bond tensor $B^\ell$ in order to update it using a gradient descent step. Because the rest of the MPS tensors are kept fixed, let us show that to compute the gradient it suffices to feed, or project, each input $\mathbf{x}_n$ through the fixed "wings" of the MPS as shown on the left-hand side of Fig. 6(b) (connected lines in the diagram indicate sums over pairs of indices). The result is a projected, four-index version of the input $\tilde{\Phi}_n$ shown on the right-hand of Fig. 6(b). The current decision function can be efficiently computed from this projected input $\tilde{\Phi}_n$ and the current bond tensor $B^\ell$ as

$$f^\ell(\mathbf{x}_n) = \sum_{\alpha_{j-1} \alpha_{j+1}} \sum_{s_j s_{j+1}} B^{\alpha_{j-1} \ell \alpha_{j+1}}_{s_j s_{j+1}} (\tilde{\Phi}_n)^{s_j s_{j+1}}_{\alpha_{j-1} \ell \alpha_{j+1}} \tag{6}$$

or as illustrated in Fig. 6(c). The gradient update to the tensor $B^\ell$ can be computed as

$$\Delta B^\ell = -\frac{\partial C}{\partial B^\ell} = \sum_{n=1}^{N_T} (y_n^\ell - f^\ell(\mathbf{x}_n)) \tilde{\Phi}_n \ . \tag{7}$$

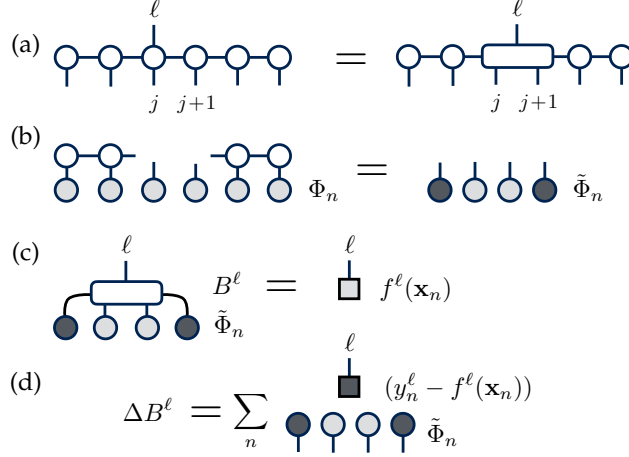

Figure 6: Steps leading to computing the gradient of the bond tensor $B^\ell$ at bond $j$: (a) forming the bond tensor; (b) projecting a training input into the "MPS basis" at bond $j$; (c) computing the decision function in terms of a projected input; (d) the gradient correction to $B^\ell$. The dark shaded circular tensors in step (b) are "effective features" formed from $m$ different linear combinations of many original features.

The tensor diagram for $\Delta B^\ell$ is shown in Fig. 6(d).

Having computed the gradient, we use it to make a small update to $B^\ell$, replacing it with $B^\ell + \eta \Delta B^\ell$ for some small $\eta$. Having obtained our improved $B^\ell$, we must decompose it back into separate MPS tensors to maintain efficiency and apply our algorithm to the next bond. Assume the next bond we want to optimize is the one to the right (bond $j + 1$). Then we can compute a singular value decomposition (SVD) of $B^\ell$, treating it as a matrix with a collective row index $(\alpha_{j-1}, s_j)$ and collective column index $(\ell, \alpha_{j+1}, s_{j+1})$ as shown in Fig. 7(a). Computing the SVD this way restores the MPS form, but with the $\ell$ index moved to the tensor on site $j + 1$. If the SVD of $B^\ell$ is given by

$$B^{\alpha_{j-1}\ell\alpha_{j+1}}_{s_j s_{j+1}} = \sum_{\alpha'_j \alpha_j} U^{\alpha_{j-1}}_{s_j \alpha'_j} S^{\alpha'_j}{}_{\alpha_j} V^{\alpha_j \ell \alpha_{j+1}}_{s_{j+1}} , \qquad (8)$$

then to proceed to the next step we define the new MPS tensor at site $j$ to be $A'_{s_j} = U_{s_j}$ and the new tensor at site $j + 1$ to be $A'^\ell_{s_{j+1}} = SV^\ell_{s_{j+1}}$ where a matrix multiplication over the suppressed $\alpha$ indices is implied. Crucially at this point, only the $m$ largest singular values in $S$ are kept and the rest are truncated (along with the corresponding columns of $U$ and $V^\dagger$) in order to control the computational cost of the algorithm. Such a truncation is guaranteed to produce an optimal approximation of the tensor $B^\ell$ (minimizes the norm of the difference before and after truncation); furthermore if all of the MPS tensors to the left and right of $B^\ell$ are formed from (possibly truncated) unitary matrices similar to the definition of $A'_{s_j}$ above, then the optimality of the truncation of $B^\ell$ applies globally to the entire MPS as well. For further background reading on these technical aspects of MPS, see Refs. [8] and [16].

Finally, when proceeding to the next bond, it would be inefficient to fully project each training input over again into the configuration in Fig. 6(b). Instead it is only necessary to advance the projection by one site using the MPS tensor set from a unitary matrix after the SVD as shown in Fig. 7(b). This allows the cost of each local step of the algorithm to remain independent of the size of the input space, making the total algorithm scale only linearly with input space size (i.e. the number of components of an input vector $\mathbf{x}$).

The above algorithm highlights a key advantage of MPS and tensor networks relevant to machine learning applications. Following the SVD of the improved bond tensor $B'^\ell$, the dimension of the new MPS bond can be chosen *adaptively* based on the number of large singular values encountered in the SVD (defined by a threshold chosen in advance). Thus the MPS form of $W^\ell$ can be compressed as much as possible, and by different amounts on each bond, while still ensuring an accurate approximation of the optimal decision function.

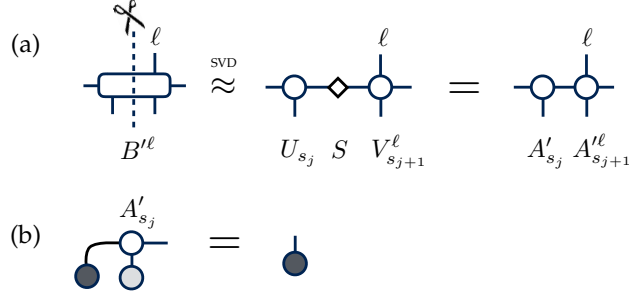

Figure 7: Restoration (a) of MPS form, and (b) advancing a projected training input before optimizing the tensors at the next bond. In diagram (a), if the label index $\ell$ was on the site $j$ tensor before forming $B^\ell$, then the operation shown moves the label to site $j+1$.

The scaling of the above algorithm is $d^3 m^3 N N_L N_T$, where recall $m$ is the typical MPS bond dimension; $N$ the number of components of input vectors $\mathbf{x}$; $N_L$ the number of labels; and $N_T$ the size of the training data set. Thus the algorithm scales linearly in the training set size: a major improvement over typical kernel-trick methods which typically scale at least as $N_T^2$ without specialized techniques [23]. This scaling assumes that the MPS bond dimension $m$ needed is independent of $N_T$, which should be satisfied once $N_T$ is a large, representative sample.

In practice, the training cost is dominated by the large size of the training set $N_T$, so it would be very desirable to reduce this cost. One solution could be to use stochastic gradient descent, but our experiments at blending this approach with the MPS sweeping algorithm did not match the accuracy of using the full, or batch gradient. Mixing stochastic gradient with MPS sweeping thus appears to be non-trivial but is a promising direction for further research.

## 5   MNIST Handwritten Digit Test

To test the tensor network approach on a realistic task, we used the MNIST data set [24]. Each image was scaled down from $28 \times 28$ to $14 \times 14$ by averaging clusters of four pixels; otherwise we performed no further modifications to the training or test sets. Working with smaller images reduced the time needed for training, with the tradeoff of having less information available for learning.

When approximating the weight tensor as an MPS, one must choose a one-dimensional ordering of the local indices $s_1, s_2, \ldots, s_N$. We chose a "zig-zag" ordering meaning the first row of pixels are mapped to the first 14 external MPS indices; the second row to the next 14 MPS indices; etc. We then mapped each grayscale image $\mathbf{x}$ to a tensor $\Phi(\mathbf{x})$ using the local map Eq. (3).

Using the sweeping algorithm in Section 4 to optimize the weights, we found the algorithm quickly converged after a few passes, or sweeps, over the MPS. Typically five or less sweeps were needed to see good convergence, with test error rates changing only hundreths of a percent thereafter.

Test error rates also decreased rapidly with the maximum MPS bond dimension $m$. For $m = 10$ we found both a training and test error of about 5%; for $m = 20$ the error dropped to only 2%. The largest bond dimension we tried was $m = 120$, where after three sweeps we obtained a test error of 0.97%; the corresponding training set error was 0.05%. MPS bond dimensions in physics applications can reach many hundreds or even thousands, so it is remarkable to see such small classification errors for only $m = 120$.

## 6   Interpreting Tensor Network Models

A natural question is which set of functions of the form $f^\ell(\mathbf{x}) = W^\ell \cdot \Phi(\mathbf{x})$ can be realized when using a tensor-product feature map $\Phi(\mathbf{x})$ of the form Eq. (2) and a tensor-network decomposition of $W^\ell$. As we will argue, the possible set of functions is quite general, but taking the tensor network structure into account provides additional insights, such as determining which features the model actually uses to perform classification.

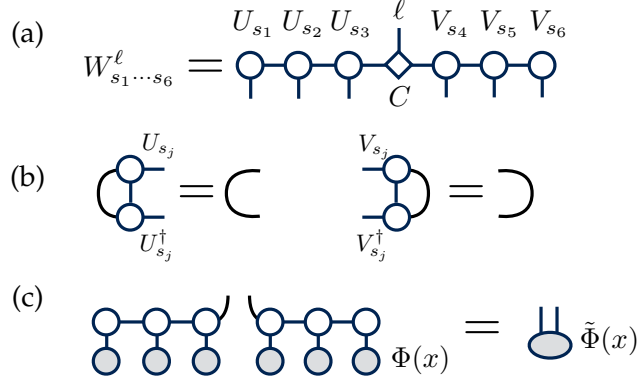

Figure 8: (a) Decomposition of $W^\ell$ as an MPS with a central tensor and orthogonal site tensors. (b) Orthogonality conditions for $U$ and $V$ type site tensors. (c) Transformation defining a reduced feature map $\tilde{\Phi}(\mathbf{x})$.

## 6.1 Representational Power

To simplify the question of which decision functions can be realized for a tensor-product feature map of the form Eq. (2), let us fix $\ell$ to a single label and omit it from the notation. We will also temporarily consider $W$ to be a completely general order-$N$ tensor with no tensor network constraint. Then $f(\mathbf{x})$ is a function of the form

$$f(\mathbf{x}) = \sum_{\{s\}} W_{s_1 s_2 \cdots s_N} \phi^{s_1}(x_1) \otimes \phi^{s_2}(x_2) \otimes \cdots \phi^{s_N}(x_N) . \tag{9}$$

If the functions $\{\phi^s(x)\}$, $s = 1, 2, \ldots, d$ form a basis for a Hilbert space of functions over $x \in [0, 1]$, then the tensor product basis $\phi^{s_1}(x_1) \otimes \phi^{s_2}(x_2) \otimes \cdots \phi^{s_N}(x_N)$ forms a basis for a Hilbert space of functions over $\mathbf{x} \in [0, 1]^{\times N}$. Moreover, in the limit that the basis $\{\phi^s(x)\}$ becomes complete, then the tensor product basis would also be complete and $f(\mathbf{x})$ could be any square integrable function; however, practically reaching this limit would eventually require prohibitively large tensor dimensions.

## 6.2 Implicit Feature Selection

Of course we have not been considering an arbitrary weight tensor $W^\ell$ but instead approximating the weight tensor as an MPS tensor network. The MPS form implies that the decision function $f^\ell(\mathbf{x})$ has interesting additional structure. One way to analyze this structure is to separate the MPS into a central tensor, or core tensor $C^{\alpha_i \ell \alpha_{i+1}}$ on some bond $i$ and constrain all MPS site tensors to be *left orthogonal* for sites $j \leq i$ or *right orthogonal* for sites $j \geq i$. This means $W^\ell$ has the decomposition

$$W^\ell_{s_1 s_2 \cdots s_N} =$$
$$\sum_{\{\alpha\}} U^{\alpha_1}_{s_1} \cdots U^{\alpha_i}_{\alpha_{i-1} s_i} C^\ell_{\alpha_i \alpha_{i+1}} V^{\alpha_{i+1}}_{s_{i+1} \alpha_{i+2}} \cdots V^{\alpha_{N-1}}_{s_N} \tag{10}$$

as illustrated in Fig. 8(a). To say the $U$ and $V$ tensors are left or right orthogonal means when viewed as matrices $U_{\alpha_{j-1} s_j}{}^{\alpha_j}$ and $V^{\alpha_{j-1}}{}_{s_j \alpha_j}$ these tensors have the property $U^\dagger U = I$ and $V V^\dagger = I$ where $I$ is the identity; these orthogonality conditions can be understood more clearly in terms of the diagrams in Fig. 8(b). Any MPS can be brought into the form Eq. (10) through an efficient sequence of tensor contractions and SVD operations similar to the steps in Fig. 7(a).

The form in Eq. (10) suggests an interpretation where the decision function $f^\ell(\mathbf{x})$ acts in three stages. First, an input $\mathbf{x}$ is mapped into the $d^N$ dimensional feature space defined by $\Phi(\mathbf{x})$, which is exponentially larger than the dimension $N$ of the input space. Next, the feature vector $\Phi$ is mapped into a much smaller $m^2$ dimensional space by contraction with all the $U$ and $V$ site tensors of the MPS. This second step defines a new feature map $\tilde{\Phi}(\mathbf{x})$ with $m^2$ components as illustrated in Fig. 8(c). Finally, $f^\ell(\mathbf{x})$ is computed by contracting $\tilde{\Phi}(\mathbf{x})$ with $C^\ell$.

To justify calling $\tilde{\Phi}(\mathbf{x})$ a feature map, it follows from the left- and right-orthogonality conditions of the $U$ and $V$ tensors of the MPS Eq. (10) that the indices $\alpha_i$ and $\alpha_{i+1}$ of the core tensor $C$ label an orthonormal basis for a subspace of the original feature space. The vector $\tilde{\Phi}(\mathbf{x})$ is the projection of $\Phi(\mathbf{x})$ into this subspace.

The above interpretation implies that training an MPS model uncovers a relatively small set of important features and simultaneously trains a decision function using only these reduced features. The feature selection step occurs when computing the SVD in Eq. (8), where any basis elements $\alpha_j$ which do not contribute meaningfully to the optimal bond tensor are discarded. (In our MNIST experiment the first and last tensors of the MPS completely factorized during training, implying they were not useful for classification as the pixels at the corners of each image were always white.) Such a picture is roughly similar to popular interpretations of simultaneously training the hidden and output layers of shallow neural network models [25]. (MPS were first proposed for learning features in Bengua et al. [4], but with a different, lower-dimensional data representation than what is used here.)

## 7 Discussion

We have introduced a framework for applying quantum-inspired tensor networks to supervised learning tasks. While using an MPS ansatz for the model parameters worked well even for the two-dimensional data in our MNIST experiment, other tensor networks such as PEPS [6], which are explicitly designed for two-dimensional systems, or MERA tensor networks [15], which have a multi-scale structure and can capture power-law correlations, may be more suitable and offer superior performance. Much work remains to determine the best tensor network for a given domain.

There is also much room to improve the optimization algorithm by incorporating standard techniques such as mini-batches, momentum, or adaptive learning rates. It would be especially interesting to investigate unsupervised techniques for initializing the tensor network. Additionally, while the tensor network parameterization of a model clearly regularizes it in the sense of reducing the number of parameters, it would be helpful to understand the consquences of this regularization for specific learning tasks. It could also be fruitful to include standard regularizations of the parameters of the tensor network, such as weight decay or $L_1$ penalties. We were surprised to find good generalization without using explicit parameter regularization.

We anticipate models incorporating tensor networks will continue be successful for quite a large variety of learning tasks because of their treatment of high-order correlations between features and their ability to be adaptively optimized. With the additional opportunities they present for interpretation of trained models due to the internal, linear tensor network structure, we believe there are many promising research directions for tensor network models.

Note: while we were preparing our final manuscript, Novikov et al. [26] published a related framework for using MPS (tensor trains) to parameterize supervised learning models.

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
