[Reviews · NeurIPS 2016]

Reviewer 1

Summary

The paper proposes the use of the matrix product states decomposition (MPS) of a tensor network for a general multiclass supervised classification problem. The paper is well written, however assumes the reader has some familiarity with the concepts of MPS and tensor networks which are commonly used techniques in the field of quantum physics. The proposed methodology builds on prior work (citations [3] and [4]) and the contribution of the paper is incremental and somehow similar in nature to the aforementioned work. The main contributions of the paper can be summarized into two main parts: a) It defines a tensor inspired encoding (kernel) that can be applied to any classification problem where the data is represented in fixed length vector form. b) It optimizes the MPS representation for the classification task at hand by using gradient descent as opposed to [4] where the MPS representation is chosen first in an unsupervised manner and then used it afterwards in a disjoint fashion for the classification task by applying a classification algorithm to the resulting transformed data.

Qualitative Assessment

1) The paper assumes some familiarity with tensor networks and MPS. This may be common in the quantum physics area but not so on the NIPS community (In my opinion). Hence the paper could be greatly improved by better introduction the reader to the concepts and build from the bottom up. This is not an easy task due to the amount of space available in a NIPS paper but nonetheless its necessary to improve readability of the paper 2) Please add dimensions to the vector and matrices. This would help improve readability as well. 3) In line 153 it is mentioned that a zig-zag ordering is used but Figure 8 does not seems to illustrate a zig-zag pattern. Is this a typo? Did you use the wrong Figure? Please clarify. 4) Please elaborate regarding the differences between the work presented here and [3] and [4] early on the paper, for a reader with few or no experience with tensor networks this information can be useful to follow the paper and understand the contributions of the work presented here. 5) Also for this audience the connection to Neural networks is very informative, however is briefly mentioned at the end of the paper. Please elaborate on this topic. 6) Can you explain how initialization is done? How sensitive the problem is to local minima? How did you deal with this problem? 7) It would be helpful to mention what tools where used to calculate SVD.

Confidence in this Review

1-Less confident (might not have understood significant parts)


Reviewer 2

Summary

This paper deals with the use of tensor networks to perform supervised learning. Specifically, the authors shows how tensor networks can be adapted to produce useful representation of data and solve classification problems. The main idea is to consider data feature maps that are built from the tensor product of local features and represent them as a higher-order tensor. A classification problem then consists in viewing the model parameters as a tensor network and learn them using a matrix-product state (MPS) decomposition. An optimization algorithm to find the MPS decomposition is described and an experiment on MNIST digit data set is given. Finally, a discussion on the class of functions in which the solution is searched is provided.

Qualitative Assessment

The idea of using tensor networks for machine learning problems is interesting. Unfortunately, there are many important issues that prevent the publication of the manuscript as it is now. My main concerns are the following: 1- Some parts of the paper are hard to read. The paper applies and adapts some idea about tensor networks that are used/developed by the Physics community to ML problems. The authors have done a good job in illustrating the idea and the intuition by many figures. However, I think that for a ML audience, giving more details on tensor networks, MPS decomposition algorithms and their use for Physics problems (in short, more details on the papers cited and published in Physics journals) will improve the readability and the accessibility of the paper. A section can be dedicated to this. 2- The notation is also heavy to follow and dos not help understanding. For example: * In equation 2, what is the relation between x and x_j? Is x_j a component of the vector x? What is N: is it the number of attribute? * In equation 3, \phi is indexed by s_j but depends only in x_j. For this example, is it the same \phi for all the s_j? * A needs to be defined in equation 5 * line 68: ‘in classifying labeled data’ should be ‘unlabeled data’ (from a given training set) * … 3- Experiments are weak. To assess the performance of the method, more data sets have to be used and not only the simple MNIST digit data. Also, some informations about the experimental setting are missing. For example, How the data set was divided in training and testing data? 4- I think that the choice of the local feature map is crucial. It is not clear how this feature map can be chosen. The one given in Eq. 3 seems to be adapted only for image data. The same for the parameters d and m. How these parameters can be chosen? 5- Is the MPS decomposition unique? If no, how this affects the learning process? 6- MPS decomposition for feature extraction and classification were used in [4]. I think that the novel contributions compared to [4] need to be clearly stated. Line 210 is too short and does not give enough details to see the difference between the two works. 7- ANOVA kernels are based on the product of a set of ‘local’ kernels. The product of the kernels allows to find a solution in a Hilbert space constructed from the tensor product of the reproducing Hilbert spaces associated to each local kernel. I Think that it is interesting to study the link between the space of functions obtained by such kernels and the tensor network representation proposed here.

Confidence in this Review

2-Confident (read it all; understood it all reasonably well)


Reviewer 3

Summary

Use of matrix product states in classification of images.

Qualitative Assessment

A very interesting and strong point of the paper is that it nicely bridges different fields. It proposes the use of tensor networks with matrix product states (also called tensor trains depending on the particular research area) for classification of images. On the other hand the presentation can be improved and clarified at some points: - section 5 "To approximate the weight tensor as an MPS": it is not clear in what sense a certain tensor is approximated or how. Do you mean that if we take all singular values in (11) then it corresponds to the tensor to be approximated? - in section 7 the authors mention some open problems about regularization. However, I think the answer is already (implicitly) present in eq (11) and (14): the eq (14) is closely related to the work of paper Signoretto M., Tran Dinh Q., De Lathauwer L., Suykens J.A.K., Learning with Tensors: a Framework Based on Convex Optimization and Spectral Regularization, Machine Learning, vol. 94, no. 3, Mar. 2014, pp. 303-351 where regularization is applied to learning of a tensor model, related to multilinear SVD, nuclear norm regularization, and singular value thresholding operations. Because in eq (11) only the m largest singular values are kept this is also influencing the rank of the tensor. In other words, the authors do not add a regularization to eq (6), but instead control the rank by the SVD truncation operations in (11). - in [15] Cichocki it is also explained how to solve a linear system by using MPS. In fact the eqs (6)(7) are also linear in the unknown tensor. Therefore it is unclear why the method proposed in [15] cannot be directly applied here. What is different from [15] or what is additionally complicating the problem in this paper? - please explain in more detail how \tilde{\Phi}_n is obtained from the given data. - eq (7)-(8): please explain what you mean by "leading-order update". - I would suggest to also add an algorithm to the paper in order to make step by step clear what needs to done to finally obtain the solution B, starting from the given training data. - eq (3): aren't these Fourier features? (note: random Fourier features are currently also used in large scale problems, not limited to images)

Confidence in this Review

2-Confident (read it all; understood it all reasonably well)


Reviewer 4

Summary

The authors approach the problem of image classification by means of tensor networks and optimization algorithms for these. A minimal theoretical evalution of their method and results on the MNIST dataset are provided.

Qualitative Assessment

I have a few problems with the paper. I would have liked to see a better embedding of the paper in the literature. For instance, classification is one of the main applications of GLMs (called logistic regression in this case) and tensor methods have been used to optimise mixtures of GLMS [1]. There is not much depth to the paper. The authors propose essentially a new algorihtm without giving the reader strong arguments for getting interested in the algorithm: classification or regression algorithms are typically compared in terms of their expressive power (dense in L2, for instance -- the authors give here an informal argument why their model class should lie dense) and in terms of rates of convergence; for most regression/classification methods minmax optimal rates are known by now and I expect some discussion of the statistical properties of a regressor. One might also motivate the algorithm by beneficial computational runtimes/ complexity compared to other approaches. But this is not done here either. Finally, a strong empirical performance might be used as a motivation and the authors provide some results on the MNIST data set. But the empirical evaluation is in total rather meager (one short section in the middle of p. 6). So the paper leaves one with the feeling of yet another heuristic to some well understood machine learning problem and one wonders why one should bother. Some detailed comments: L19: what do you mean by exponentially large; exponential in what? L95: "The cost function for which we found our best test results .."; what does that mean? The performance is defined in terms of a cost function so what do you mean by your best test results?? Eq 6: how is delta^ell_L_n defined? == 1 if L_n = ell? L168: What do you mean by "the possible set of functions is quite general, .. " ? L176: a basis is by definition complete. What you mean is an orthonormal system. Literature should be alphabetical. [1] Provable Tensor Methods for Learning Mixtures of Generalized Linear Models; Sedghi et al.; AISTATS 2016.

Confidence in this Review

2-Confident (read it all; understood it all reasonably well)


Reviewer 5

Summary

This paper investigates tensor networks from quantum physics in order to provide a supervised learning algorithm.

Qualitative Assessment

A major drawback of this work is that it is not motivated, in the sense of using the tensor networks for the definition of the feature map. It is clear that the use of tensor networks leads to several difficulties in the optimization process. We recommend the authors to study the possible use of the kernel trick, which could alleviate the computational complexity, as it has already been largely investigated in kernel machines, such as support vector machines. Within the proposed classification model as given in Section 3, it is difficult to understand the issue of the label, namely how the label index is arbitrarily placed at some position of the tensor, and how/why it can be moved to another tensor. This could be viewed as affecting the label to a single pixel of the image, and not to the whole image. This is due to the considered model (4)-(5) which seems to be not appropriate for multiclass classification. The authors used the quadratic loss in (6), motivated by having the best test results. They should also state the other used loss functions in the comparaison. It is worth noting that the quadratic loss provide relatively easy computations, as opposed to some other loss functions, due to its derivability. There are many issues regarding the experiments : -Several details are missing, such as the number of data used in the experiments, i.e., N, and the number of classes, i.e., L. -The 28x28 images are subsampled to 14x14, in order to reduce the training time. However, nothing is said about the computational complexity of the algorithm. This is a major issue to be addressed in the paper. -It is not clear how the training and test errors are estimated. -The proposed classification algorithm is not compared to any of the state-of-the-art classification methods. It should be compared in terms of the estimated errors, as well as the computational complexity.

Confidence in this Review

1-Less confident (might not have understood significant parts)


Reviewer 6

Summary

This paper develops an efficient algorithm for classifying images. The authors demonstrate how to adapt the tensor networks optimization algorithms to supervised learning tasks by using matrix product states. The authors first map the very large vectors X into a higher dimensional space via a feature map Φ(x), and then use a decision function f(x)=W×Φ(x) to classify these vectors. Because the feature vector Φ(x) and the weight vector W can be exponentially large or even infinite, the authors proposed a new approach which is different from kernel trick (reference 9). This approach uses a tensor network to approximate the optimal weight vector W (Eq.5) , by extracting information hidden within trained model and exploiting the structure of W, a “sweeping” optimization algorithm is developed (Eq.7) to minimize a quadratic cost function defining the classification task.

Qualitative Assessment

This paper develops an efficient algorithm for classifying images. The authors demonstrate how to adapt the tensor networks optimization algorithms to supervised learning tasks by using matrix product states. The authors first map the very large vectors X into a higher dimensional space via a feature map Φ(x), and then use a decision function f(x)=W×Φ(x) to classify these vectors. Because the feature vector Φ(x) and the weight vector W can be exponentially large or even infinite, the authors proposed a new approach which is different from kernel trick (reference 9). This approach uses a tensor network to approximate the optimal weight vector W (Eq.5) , by extracting information hidden within trained model and exploiting the structure of W, a “sweeping” optimization algorithm is developed (Eq.7) to minimize a quadratic cost function defining the classification task. Overall, I think this paper is well organized and clearly written; the proposed approach is novel and makes an important contribution to the field of machine learning. It is nice to build a connection between tensor networks and feature selection. However, one major problem of this paper is the experiment part. To demonstrate the effectiveness of the proposed approach, the authors uses one single dataset, i.e., MINST, and achieves less than 1% test set classification error. It would be better if the authors can apply the proposed approach to other widely used dataset in this field and show the results. More importantly, there is a lack of comparison with state-of-the-art learning algorithms and feature selection methods, making it difficult to judge the performance of the proposed approach. I hope the authors can clarify these questions in the rebuttal process. Overall, my opinion for this paper is weak accept.

Confidence in this Review

2-Confident (read it all; understood it all reasonably well)